# Electrocardiography Classification with Leaky Integrate-and-Fire Neurons in an Artificial Neural Network-Inspired Spiking Neural Network Framework

**DOI:** 10.3390/s24113426

**Published:** 2024-05-26

**Authors:** Amrita Rana, Kyung Ki Kim

**Affiliations:** Department of Electronic Engineering, Daegu University, Daegudaero 201, Gyeongsan 38543, Republic of Korea; amritaranamagar60@gmail.com

**Keywords:** ECG classification, spiking neural network, leaky integrate-and-fire, ANN-to-SNN conversion

## Abstract

Monitoring heart conditions through electrocardiography (ECG) has been the cornerstone of identifying cardiac irregularities. Cardiologists often rely on a detailed analysis of ECG recordings to pinpoint deviations that are indicative of heart anomalies. This traditional method, while effective, demands significant expertise and is susceptible to inaccuracies due to its manual nature. In the realm of computational analysis, Artificial Neural Networks (ANNs) have gained prominence across various domains, which can be attributed to their superior analytical capabilities. Conversely, Spiking Neural Networks (SNNs), which mimic the neural activity of the brain more closely through impulse-based processing, have not seen widespread adoption. The challenge lies primarily in the complexity of their training methodologies. Despite this, SNNs offer a promising avenue for energy-efficient computational models capable of displaying a high-level performance. This paper introduces an innovative approach employing SNNs augmented with an attention mechanism to enhance feature recognition in ECG signals. By leveraging the inherent efficiency of SNNs, coupled with the precision of attention modules, this model aims to refine the analysis of cardiac signals. The novel aspect of our methodology involves adapting the learned parameters from ANNs to SNNs using leaky integrate-and-fire (LIF) neurons. This transfer learning strategy not only capitalizes on the strengths of both neural network models but also addresses the training challenges associated with SNNs. The proposed method is evaluated through extensive experiments on two publicly available benchmark ECG datasets. The results show that our model achieves an overall accuracy of 93.8% on the MIT-BIH Arrhythmia dataset and 85.8% on the 2017 PhysioNet Challenge dataset. This advancement underscores the potential of SNNs in the field of medical diagnostics, offering a path towards more accurate, efficient, and less resource-intensive analyses of heart diseases.

## 1. Introduction

As reported by the World Health Organization, cardiovascular diseases (CVDs) represent the foremost cause of mortality worldwide [1]. In response, a myriad of strategies and actions have been initiated across various sectors that are aimed at diminishing the frequency of cardiovascular incidents. Within this context, the electrocardiogram (ECG) has emerged as a pivotal tool for the immediate identification of cardiac conditions. Serving as a visual chart of the heart’s electrical dynamics, the ECG plays a critical role in pinpointing a spectrum of cardiac irregularities [2]. Conventional works include ECG monitoring solutions that extract the R-peak or the QRS complex and do not further process the ECG signal [3,4]. Recent research has predominantly targeted the augmentations ECG-based diagnostics, notably in the accurate delineation of the QRS complex, as depicted through the Q, R, and S wave patterns illustrated in Figure 1. This focus underscores the significance of ECGs in advancing cardiovascular health diagnostics and intervention strategies.

However, classical algorithms were mainly based on morphological features and are not sufficient for accurate arrhythmia detection [5].

The quest for enhanced arrhythmia detection has led to a significant investment in the development of sophisticated neural network architectures, including Deep Convolutional Neural Networks (CNNs) and Recurrent Neural Networks (RNNs) [6,7,8,9]. Despite their proven efficacy, these technological solutions bring forth increased hardware expenses, operational complexities, heightened power consumption, and prolonged deployment durations.

Having emerged as a contrasting paradigm, Spiking Neural Networks (SNNs) adopt a binary, event-driven operation mode that mirrors the activity of biological neurons, which are activated exclusively in response to incoming or outgoing neural spikes. This operational characteristic endows SNNs with a remarkable energy efficiency and reduces their dependency on the substantial computational demands that are typical of their convolutional counterparts. SNNs have gained traction in a variety of domains, including medical imaging and ECG signal analysis, thus reinforcing their burgeoning status within the sphere of neuroscience research [10,11,12].

This paper presents a cutting-edge SNN architecture inspired by a deep ANN, that integrates leaky integrate-and-fire (LIF) neurons and has been engineered to boost the efficiency and effectiveness of ECG signal classification. This initiative rectifies the limitations of previous SNN deployments, which often overlooked the complexity of ECG signals, potentially undermining diagnostic precision [13,14]. Previous SNN-based research processed ECG signals by employing non-SNN algorithms, neglecting full signal analysis, and applying bandpass or perturbation detection [15,16,17,18]. Our proposed model incorporates an attention mechanism within the SNN framework, enhancing cardiac feature extraction through the meticulous analysis of channel information and the identification of unique patterns. The necessity of this approach becomes apparent when considering the significant variation and similarities observed among periodic ECG signal categories, as illustrated in Figure 2.

## 2. Proposed Method

In this paper, we introduce an ANN-inspired SNN framework with an attention mechanism to automate the extraction of distinctive representations learned from ECG signals for precise recognition of abnormal conditions.

### 2.1. Denoising ECG Signals

The wavelet thresholding method stands out for its effectiveness in purifying ECG signals by removing extraneous noise without compromising the integrity of the waveform’s morphological characteristics [19]. This technique employs a strategic application of wavelet transform methods to differentiate between signal components attributable to cardiac activity and those arising from noise. By setting an optimal threshold level, it is possible to attenuate noise components, thus enhancing the clarity and quality of the ECG signal. This preparatory step is crucial for ensuring that subsequent analyses are based on data that most accurately reflect the underlying cardiac dynamics.

### 2.2. ECG Segmentation

Segmentation plays a pivotal role in the analysis of ECG signals, particularly in the context of identifying and examining individual heartbeats within a continuous ECG recording, as depicted in Figure 2. Our approach draws upon the rich annotations provided by the MIT-BIH dataset, which comprises 44 records, to pinpoint the R-wave peak—the most distinctive feature of each heartbeat [20]. Recognizing the potential for slight discrepancies between the annotated peak values and those derived post-denoising, our methodology includes a provision for adjusting these values to represent the true peak positions more accurately.

In recognizing the importance of analyzing heartbeats that reflect a physiologically stable state, our protocol excludes the initial fifteen heartbeats from each sequence. This decision is based on the premise that the beginning of each recording may capture transient physiological states that are not representative of the subject’s typical cardiac function.

To enable a detailed comparison of heartbeats and facilitate the extraction of diagnostic features, we adopt a systematic approach to extract segments surrounding the R-wave peak. By delineating a window that extends 250 points ahead and 250 points behind each R-wave peak, we ensure the inclusion of the entire P, QRS, and T wave components within each segmented heartbeat. This segmentation strategy not only standardizes the length of the feature vectors for subsequent analysis but also ensures that the critical components of each heartbeat are comprehensively captured, laying the groundwork for a more nuanced and precise interpretation of the ECG data.

### 2.3. SNN

Building upon the innovative foundation of the 12-layer Artificial Neural Network (ANN) model, our proposed method introduces a groundbreaking Spiking Neural Network (SNN) architecture, delineated in Figure 3, which deftly incorporates an attention mechanism within its structure.

#### 2.3.1. Leaky Integrate-and-Fire

The leaky integrate-and-fire (LIF) model is esteemed due to its biological plausibility and computational efficacy, as demonstrated in Figure 4. Recently, researchers have designed an SNN consisting of LIF neurons that can be used to detect anomalous patterns in ECGs [21]. The dynamism of the LIF neuron is underpinned by its membrane potential’s sensitivity to incoming signals—each contributing to the temporal build-up of the neuron’s potential until a threshold *V_th_* is surpassed, resulting in the emission of a spike.

The input spikes, as shown on the top graph of Figure 4, initiate the response of the LIF neuron. The synaptic weight *w_ij_* modulates the influence of each incoming spike from presynaptic neuron *i* onto postsynaptic neuron *j*. This modulation is pivotal to the network’s learning and pattern recognition capabilities. The middle graph illustrates the neuron’s membrane potential over time. It accumulates charge in response to the input spikes, decays over time due to its ‘leaky’ nature, and resets after spiking back to the resting potential *V_r_*, which is indicative of the neuron’s refractory period. The output spikes, depicted on the bottom graph, are the neuron’s response to reaching the critical threshold *V_th_*. This spike, occurring at a discrete point in time, underscores the neuron’s role as an information-processing unit in an SNN.

In the SNN layer, the membrane potential of the LIF neuron is ingeniously engineered to parallel the original activation function served by the convolution operation in traditional convolutional neural networks.

Equation (1) provides a mathematical framework for quantifying the temporal information storage capacity of $U_{j}$(*t*), offering insights into the potential’s evolution and its implications for signal processing within the network.
(1)$$\tau_{dt}^{{dU}_{j}(t)}=-U_{j}(t)+RI(t)$$
where τ indicates the time constant for the leak current, and *R* and *I*(*t*) represent the input resistance and drive current in the LIF circuit, respectively. The membrane voltage is represented by Equation (2).
(2)$$U_{j}\left(t\right)=\lambda U_{j}\left(t-1\right)+\sum_{i}w_{ij}U_{i}(t)$$
where $U_{j}$(*t*) is the membrane voltage of the postsynaptic neuron j at the time *t* and $w_{ij}\;$is the synaptic weight which weights the inputs of presynaptic neuron i. An action potential is triggered and a spike is produced by neuron j when the membrane voltage $V_{m}$(*v*) surpasses the voltage threshold *Vth*. One can compute the output spike at time *t* as follows:(3)$$\begin{array}{cc} Output\;\left(t\right)=1, & {\;if\;U}_{j}\left(t\right)>Vth\\ 0, & otherwise \end{array}$$

After the neuron *j* attains its threshold, the membrane potential $U_{j}$(*t*) is reset to the predefined reset potential $V_{r}$. As a simplifying assumption in our model, $V_{r}$ is designated as zero. This reset initiates the refractory period for neuron *j*, a span during which the membrane potential is held constant, rendering the neuron temporarily unresponsive to any new stimuli. The initiation of the refractory period simulates the biological neuron’s recovery phase, where it is not capable of firing another action potential regardless of the incoming signals.

#### 2.3.2. Attention Module

As shown in Figure 5, the proposed attention module is stacked by an input layer that receives the ECG signal. The LIF neurons, embedded at key intervals in conjunction with the attention mechanism, replace the traditional ReLU activation functions, marking a significant shift toward an architecture that closely emulates the complex operations between biological neural systems. The attention layer consists of 2 CONV followed by 2 POOL layers, which use 2 kernels in the first and second CONV that increase the depth to 2. The attention layer is then followed by 4 SNN layers.

This strategic configuration facilitates a more nuanced response to the spatiotemporal aspects of the input signals. The attention layer is applied to the SNN layer to measure the importance of the input sample and takes control of how much attention the model should pay to each signal slice. To be specific, a weight is assigned to each time slice in the input ECG sample. For an input sample with 250 time slices, we use an FC layer to learn an attention–weight vector with 250 elements. A SoftMax is used to normalize to make the sum equal to 1. Thereafter, we broadcast the weights to the same shape and conduct element-wise multiplication to adjust the dimension. We then incorporate multi-head attention to independently calculate multiple attention weights. Finally, averaging all the attention heads results in the final attention weights. The notation for the single heartbeat is shown in Equation (4).
(4)$$X_{i}=€R\left[n_{i},K_{i},d_{i}\right]$$
where n_i_, K_i,_ and d_i_ denote the number of channels and the dimension of the representation depth of the *i*-th layer.

## 3. Results and Discussion

In the pursuit of enhancing arrhythmia detection through the use of Spiking Neural Networks (SNNs), this paper carries out a comprehensive examination of SNN configurations, employing a diverse ECG dataset for model training and validation, and investigating the computational intricacies of SNN optimization.

### 3.1. Dataset Utilization and Computational Framework

Two publicly available datasets, namely the MIT-BIH Arrhythmia dataset, and the 2017 PhysioNet Challenge, were used for evaluation purposes [22,23]. The computational framework used to optimize the translated SNN model operates within the Spikingjelly environment, an extension of the PyTorch library [24]. This environment offers four principal modules that are essential to the deep learning processes of SNNs, labeled as neuron, layer, functional, and encoding. Within the SNN paradigm, LIF neurons from the neuron module interpret each data input as a series of binary spikes dispersed over discrete timesteps (ranging from one to T). The evaluation metrics for ECG classification performance in our paper are accuracy, precision, and F1 score. In the case of the MIT-BIH Arrhythmia dataset, the training and testing consist of 44 recordings from 100 to 234, excluding (102, 104, 107, 217). Five major groups (N, S, V, F, and Q) of heartbeats were taken. The PhysioNet 2017 dataset provides ECG recordings (between 30 s and 60 s in length) in which the recording shows four classes i.e., normal sinus rhythm, atrial fibrillation (AF), an alternative rhythm, and noisy. Table 1 shows the performance evaluation over all the classes of PhysioNet 2017 and MIT-BIH Arrhythmia Dataset.

### 3.2. Performance Analysis

Within this study, we conducted ablation experiments on various model configurations: a twelve-layer ANN model and a six-layer SNN model. Notably, two activation functions—ReLU and Leaky ReLU—were subjected to comparative simulations, the results of which are visually presented in Figure 6a,b. The efficacy of these activation functions in conjunction with ANN layers was quantitatively assessed, with a focus on precision metrics during the classification of ECG signals. The best performance we achieved was with the LeakyReLU activation in the ANN. The empirical findings show that the proposed SNN outperformed the 12-layer ANN by more than 1% in the case of PhysioNet, whereas it outperformed the ANN by 4% on the MIT-BIH dataset. The simulations that informed these findings were executed within the PyCharm Integrated Development Environment (IDE), harnessing the computational framework provided by PyTorch. A singular NVIDIA RTX 3090 GPU was deployed for this task, underscoring the efficiency of the computational process. Adhering to a “70:30” split strategy, the heartbeat data from all 44 ECG recordings were randomly assigned to constitute a training set (70%) and a testing set (30%), fostering a comprehensive learning and validation cycle.

The training regimen encompassed 200 epochs executed with a learning rate set at 0.001 and a batch size of 64. An initial timestep of 100 was specified to kickstart the simulation process. Remarkably, the entire simulation—encompassing the classification of ECG signals—was completed within a span of merely 2.5 h, demonstrating not only the methodological efficiency but also the computational speed of the proposed SNN model.

We then compared the performance of the model on the normal and noise classes in PhysioNet 2017, as shown in Table 2. The proposed att-SNN outperformed the other model in both cases by 4% and 0.4%.

For a fair evaluation, while comparing the model with other state-of-the-art methods, we adopted two strategies, namely the “70:30” strategy and the Inter-Patient strategy. The “70:30” strategy means that all the heartbeats from 44 ECG recordings are split with a random distribution between a training (70%) and a testing (30%) set. The overall performance of the model in identifying five types of heartbeats and four in the case of PhysioNet 2017 was far better than that of the 12-layer ANN by 85.8% and 93.8%, respectively.

Crating similar proportions for each type of heartbeat while conforming to the Inter-Patient lead to the training set containing 22 ECG records (101, 106, 108, 109, 112, 114, 115, 116, 118, 119, 122, 124, 201, 203, 205, 207, 208, 209, 215, 220, 223, and 230) and the testing containing 100, 103, 105, 111, 113, 117, 121, 123, 200, 202, 210, 212, 213, 214, 219, 221, 222, 228, 231, 232, 233, and 234. Despite having the highest power consumption, the F1 score of the V class was the highest.

We compared our model with a number of baseline models, shown in Table 3, for the MIT-BIH Arrhythmia dataset. More specifically, our overall classification accuracy was 3% higher than [25,26] and 1% higher than [27]. The F1 score is the highest among the baseline models. We tested our model in FPGA using Vivado 2019.2. However, the power consumption was the highest. Our model achieved the best accuracy with five timesteps. Figure 7 shows the comparison of different models concerning the timesteps (T).

As indicated in Figure 7 our proposed attention mechanism has an overall comparable performance in terms of classification accuracy.

Figure 7 provides a comparative analysis of accuracy trends over varying durations, measured in timesteps, between an Artificial Neural Network (ANN) a Spiking Neural Network (SNN), and three other baseline models. This phenomenon can be attributed to the networks’ increased ability to integrate and process information over extended temporal sequences, allowing for more robust signal analysis and pattern recognition.

In this paper, a twelve-layer ANN was adeptly transformed into a six-layer SNN through the integration of an attention module that employs leaky integrate-and-fire (LIF) neurons. This transformation was carried out to ensure the preservation of accuracy while leveraging the inherent temporal processing advantages of SNNs. Notably, the empirical results illustrated the effectiveness and feasibility of our classification model which achieved comparable results to previous conventional models except for its power consumption.

## 4. Conclusions

In this paper, we have pioneered an attention-augmented Spiking Neural Network (SNN) that enhances the efficacy of conventional Artificial Neural Network (ANN) frameworks for ECG signal classification. A comprehensive suite of experiments was conducted to ascertain the most effective network structures and to evaluate the impact of various activation functions on model performance. The empirical evidence gleaned from these investigations confirms that our ANN-to-SNN conversion, empowered by the integration of an attention mechanism, yields superior classification outcomes compared to the original ANN model.

Looking ahead, our aspiration is to refine this algorithm further, optimizing it for deployment on portable devices. The goal is to achieve a level of computational efficiency that allows for real-time, on-the-go cardiac monitoring, thereby contributing to proactive health management and potentially life-saving interventions. This future work aims to reduce power consumption and bridge the gap between advanced cardiac diagnostic tools and their accessibility in the form of handheld technology.

## Figures and Tables

**Figure 1 sensors-24-03426-f001:**
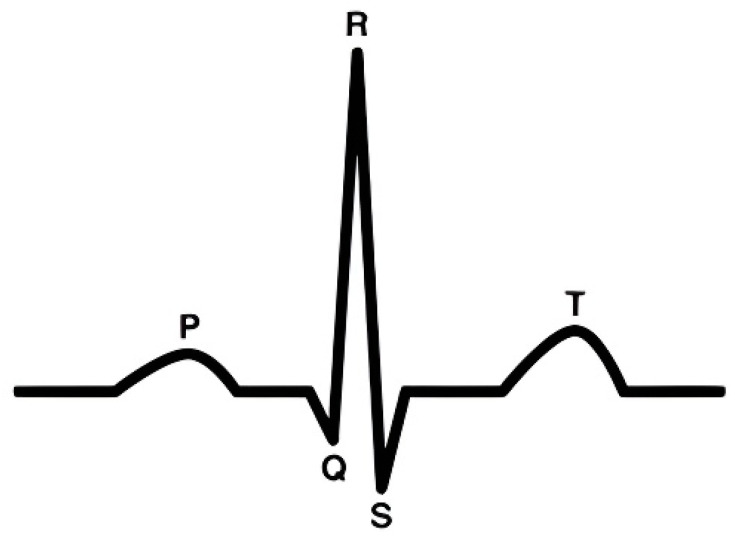
Structure of an ECG signal.

**Figure 2 sensors-24-03426-f002:**
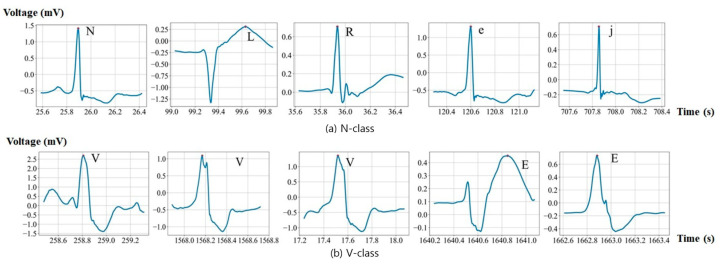
Segmented heartbeats of N and V classes from MIT-BIH Arrhythmia dataset.

**Figure 3 sensors-24-03426-f003:**
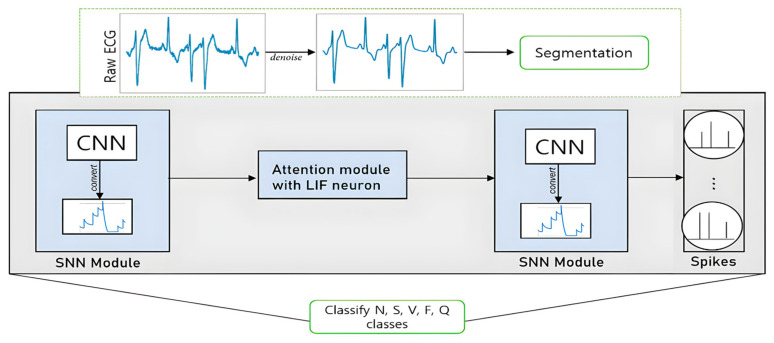
The proposed SNN framework for ECG classification.

**Figure 4 sensors-24-03426-f004:**
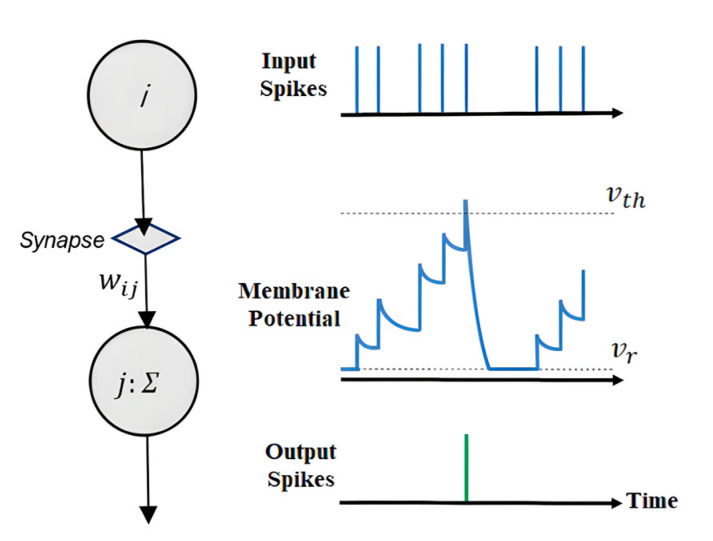
LIF neuron.

**Figure 5 sensors-24-03426-f005:**
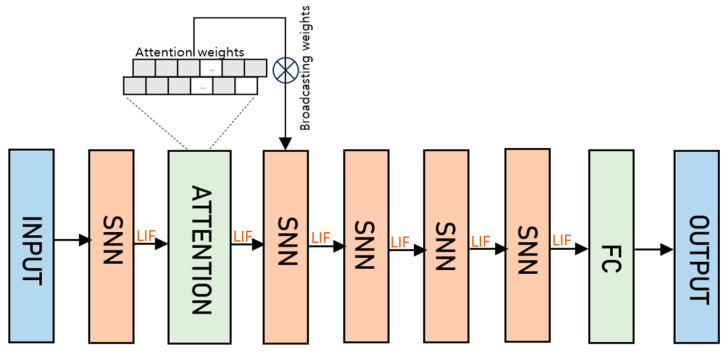
Attention Module.

**Figure 6 sensors-24-03426-f006:**
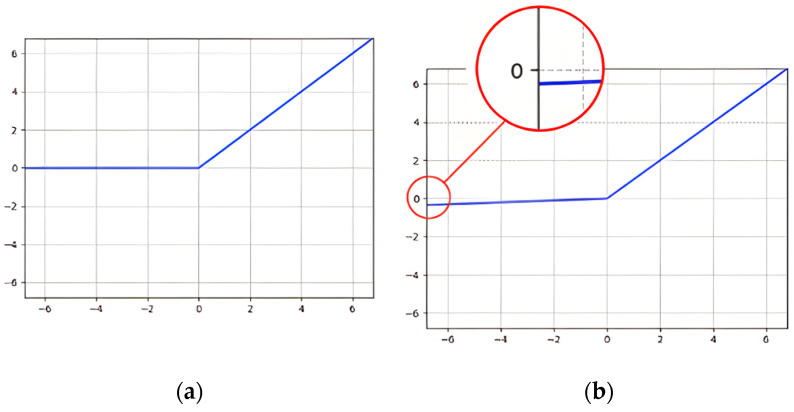
(**a**) ReLU; (**b**) Leaky ReLU.

**Figure 7 sensors-24-03426-f007:**
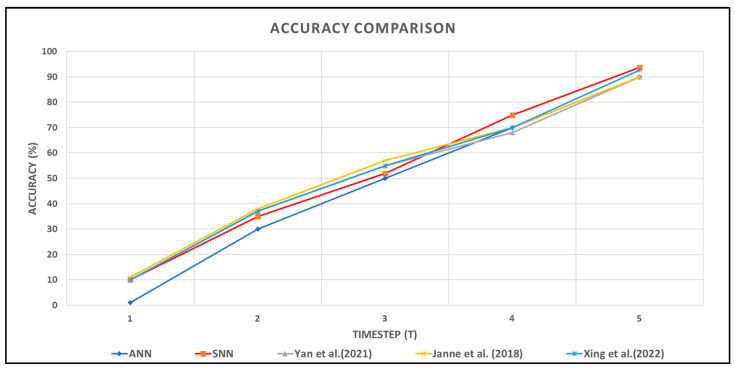
Accuracy comparison of conventional vs. proposed SNN over time steps [25,26,27].

**Table 1 sensors-24-03426-t001:** Performance evaluation over all classes.

Model	Overall-Acc (%)	F1 (%)	Pre (%)
12-layer ANN (*PhysioNet2017*)	84.1	70.1	73.4
**Proposed SNN** (*PhysioNet2017*)	**85.8**	70.4	75.7
12-layer ANN (*MIT-BIH Arrhythmia*)	89.9	83.6	85.0
**Proposed SNN** (*MIT-BIH Arrhythmia*)	**93.8**	85.4	88.0

**Table 2 sensors-24-03426-t002:** Performance evaluation over normal and noise classes.

	Model	Overall-Acc (%)	F1 (%)	Pre (%)
**Normal**	ANN	86.8	90.0	89.6
	**Proposed SNN**	**90.8**	92.0	93
**Noise**	ANN	93.4	49.1	60.0
	**Proposed SNN**	**93.0**	50.0	54.0

**Table 3 sensors-24-03426-t003:** Baseline model and proposed Att-SNN for Inter-Patient evaluation strategy.

Model	Acc (%)	Sen (%)	Pre (%)	F1 (%)	Power (W)
**Proposed Att-SNN**	**93.8**	N: 94	N: 94.3	N: 95.5	278 mW
		V: 54	V: 75.0	V: 73.5	
Yan et al. [25]	90	N: 92	N: 97	N: 94.4	181 mW
		V: 77	V: 59	V: 66.8	
Janne et al. [26]	90	N: 92	N: 97	N: 93.4	-
		V: 89	V: 51	V: 63	
Xing et al. [27]	92.07	N: 98.3	N: 93.2	N: 95.7	246 mW
		V: 69.0	V: 76.4	V: 72.5	

## Data Availability

Data are contained within the article.

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
