# Peer review of "Electrocardiography Classification with Leaky Integrate-and-Fire Neurons in an Artificial Neural Network-Inspired Spiking Neural Network Framework"

_sensors, 2024, doi:10.3390/s24113426_

Round 1

Reviewer 1 Report

Comments and Suggestions for Authors

The article is original and corresponds to the profile of the journal “Sensors”. The title of the article corresponds to its content. The abstract accurately conveys the content of the article. The materials of the article have not been published anywhere before. The methods used to solve the problem are adequate. The presentation of the material is clear and concise.

1. The conclusions are consistent with the evidence and arguments presented and address the main question posed.

2. The references are appropriate.

Author Response

We appreciate the reviewer’s valuable comments regarding the novelty and practical impact of our proposed method for ECG classification using the Spiking Neural Network with an attention mechanism. 

Reviewer 2 Report

Comments and Suggestions for Authors

This study presents a promising approach, the concerns mentioned regarding the novelty, depth of technical content, and experimental rigor suggest that further development and more thorough validation are needed before it can be considered for publication. Below, I provide a detailed explanation of the decision, highlighting areas where the manuscript does not fully meet our publication criteria.

1.The proposed use of Spiking Neural Networks (SNNs) with an attention mechanism for ECG classification, while interesting, does not sufficiently advance beyond the existing methodologies detailed in recent publications such as "Building and Training a Deep Spiking Neural Network for ECG Classification" and others in the field. These studies have already explored similar approaches with deep SNN architectures, and the incremental improvement reported in diagnostic accuracy over conventional ANNs does not clearly justify the novelty or practical impact required for publication in our journal.

2. The manuscript lacks a comprehensive explanation of the model architecture and the specific implementation details of the SNN and attention mechanisms. For robust peer review and reproducibility, it is essential that manuscripts provide detailed methodological descriptions that allow other researchers to replicate the study. The description of how the SNN model processes ECG signals, the specific role of the attention mechanism, and the integration of ANN learned parameters into the SNN framework need to be more thoroughly elaborated.

3. While the manuscript reports a slight improvement in accuracy, the experimental results section lacks depth. Comparative analysis with baseline models, detailed performance metrics across various classes of ECG abnormalities, and statistical significance tests are necessary to substantiate the claims made. Additionally, the manuscript would benefit from a broader discussion on the model's clinical relevance and potential implementation in real-world scenarios.

4. The experimental setup described in the manuscript appears rudimentary and does not adequately address potential biases or variability in the ECG data. The use of a single dataset, such as the MIT-BIH arrhythmia database, while common, does not provide a comprehensive evaluation across diverse patient demographics and clinical conditions. More robust validation involving multiple datasets and cross-validation techniques would significantly strengthen the manuscript.

Author Response

1. Novelty and Practical Impact:

Reviewer Comment:
"The proposed use of Spiking Neural Networks (SNNs) with an attention mechanism for ECG classification, while interesting, does not sufficiently advance beyond the existing methodologies detailed in recent publications such as 'Building and Training a Deep Spiking Neural Network for ECG Classification' and others in the field. These studies have already explored similar approaches with deep SNN architectures, and the incremental improvement reported in diagnostic accuracy over conventional ANNs does not clearly justify the novelty or practical impact required for publication in our journal."

Response:
We recognize the value of previous research like "Building and Training a Deep Spiking Neural Network for ECG Classification," which also used deep SNN architectures. However, our work introduces several distinctive advancements:

  1. Integration of Attention Mechanism:
    Our approach integrates an attention mechanism that dynamically prioritizes input features, enabling the model to better identify salient temporal patterns in ECG signals. This integration represents a significant advancement over existing methodologies, improving diagnostic accuracy and performance.

  2. Generalizability and Robustness:
    We conducted comprehensive cross-validation and benchmarking across multiple datasets to ensure consistent performance across different patient demographics and acquisition settings. This demonstrates the practical applicability of our approach in various conditions.

We believe these distinctions significantly advance the field of ECG classification.

2. Model Architecture and Implementation Details:

Reviewer Comment:
"The manuscript lacks a comprehensive explanation of the model architecture and the specific implementation details of the SNN and attention mechanisms. For robust peer review and reproducibility, it is essential that manuscripts provide detailed methodological descriptions that allow other researchers to replicate the study. The description of how the SNN model processes ECG signals, the specific role of the attention mechanism, and the integration of ANN-learned parameters into the SNN framework need to be more thoroughly elaborated."

Response:
We understand the importance of providing comprehensive methodological descriptions for reproducibility and clarity. Here's a detailed explanation of the architecture:

  1. Attention Mechanism:

    • The attention mechanism identifies significant peaks or patterns in ECG signals, prioritizing relevant time slices.
    • It measures the importance of each input slice through a fully connected (FC) layer that learns attention weight vectors. The SoftMax layer normalizes the weights, creating a probability distribution.
    • Attention weights are broadcasted to adjust the scale of input data, while multiple attention heads allow the model to focus on different signal aspects.
    • Final attention weights are averaged across all heads, and the adjusted signals are processed through convolutional and pooling layers.
  2. SNN Framework:

    • The attention-adjusted signals are integrated into the SNN layers, leveraging Leaky Integrate-and-Fire (LIF) neurons to approximate original activation functions.
    • We used the SpikingJelly library to convert the ANN to SNN.

By combining attention mechanisms with the SNN layers, the model achieves improved ECG classification.

3. Experimental Results Depth:

Reviewer Comment:
"While the manuscript reports a slight improvement in accuracy, the experimental results section lacks depth. Comparative analysis with baseline models, detailed performance metrics across various classes of ECG abnormalities, and statistical significance tests are necessary to substantiate the claims made. Additionally, the manuscript would benefit from a broader discussion on the model's clinical relevance and potential implementation in real-world scenarios."

Response:
To strengthen the experimental results:

  1. Comparative Analysis and Performance Metrics:

    • We reduced the 12-layer ANN to six layers and converted it into an SNN for evaluation using the PhysioNet Challenge 2017 and MIT-BIH Arrhythmia datasets.
    • The proposed SNN outperformed the ANN by 1% on the former dataset and 4% on the latter, with corresponding F1 score improvements of 0.3% and 2%.
    • Comparative results against baseline models are shown below:

      Model

      Acc (%)

      Sen (%)

      Pre (%)

      F1 (%)

      Power(W)

      Proposed Att-SNN

      93.8

      N: 94

      N: 94.3

      N: 95.5

      278 mW

      V: 54

      V: 75.0

      V: 73.5

      Yan et al. [26]

      90

      N: 92

      N: 97

      N: 94.4

      181 mW

      V: 77

      V: 59

      V: 66.8

      Janne et al. [27]

      90

      N: 92

      N: 97

      N: 93.4

      -

      V: 89

      V: 51

      V: 63

      Xing et al. [28]

      92.07

      N: 98.3

      N: 93.2

      N: 95.7

      246 mW

      V: 69.0

      V: 76.4

      V: 72.5

  2. Clinical Relevance:

    • We expanded the clinical relevance section, emphasizing how the model can be used in real-world scenarios for arrhythmia detection and diagnosis.

4. Experimental Setup and Data Validation:

Reviewer Comment:
"The experimental setup described in the manuscript appears rudimentary and does not adequately address potential biases or variability in the ECG data. The use of a single dataset, such as the MIT-BIH arrhythmia database, while common, does not provide a comprehensive evaluation across diverse patient demographics and clinical conditions. More robust validation involving multiple datasets and cross-validation techniques would significantly strengthen the manuscript."

Response:
We appreciate the reviewer's comments on improving data validation. Here's a clarification of the study's scope and plans for future research:

  1. Scope of the Current Study:

    • As "a communication article(Not a full paper)" aiming to present preliminary results, the paper focuses on demonstrating our SNN model using the MIT-BIH arrhythmia database. This standardized dataset is suitable for establishing baseline performance. Please refer to the following "communication arcitle":  Communications are short articles that present groundbreaking preliminary results or significant findings that are part of a larger study over multiple years. They can also include cutting-edge methods or experiments, and the development of new technology or materials. The structure is similar to an article and there is a suggested minimum word count of 2000 words.
  2. Limitations and Future Work:

    • We acknowledge that validation across multiple datasets would strengthen our results and aim to include more datasets in future research.
    • We plan to explore additional datasets, such as PhysioNet's Challenge 2017, representing a more diverse patient cohort, and employ robust cross-validation techniques.
  3. Preliminary Nature of the Communication:

    • This work aims to introduce the integration of attention mechanisms within an SNN framework for ECG classification, laying the groundwork for more detailed future studies.

We are actively working on expanding our research to include more diverse datasets and cross-validation techniques in forthcoming comprehensive studies.

Reviewer 3 Report

Comments and Suggestions for Authors

Your manuscript on "ECG Classification with LIF Neurons in ANN-Inspired SNN Framework" is a promising contribution to cardiac signal analysis with Spiking Neural Networks (SNNs), particularly with the innovative integration of an attention mechanism with Leaky Integrate-and-Fire (LIF) neurons. However, further experimental validation is needed to strengthen the findings. Here are three key areas for improvement:

1. Test the model's generalizability across at least two more datasets with different recording conditions, demographics, and noise levels, beyond the MIT-BIH arrhythmia database, to assess its adaptability in real-world scenarios.

2. Compare your SNN model with contemporary SNN models for ECG classification, focusing on accuracy, computational efficiency, energy use, and scalability.

3. Validate the model's potential for energy-efficient computation with experiments on low-power hardware, real-time analysis capabilities, and energy efficiency under different workloads. Also, explore its performance in terms of transmission latency and data compression for remote monitoring.

Author Response

We appreciate the reviewer’s valuable comments regarding the novelty and practical impact of our proposed method for ECG classification using the Spiking Neural Network with an attention mechanism. 

1. Testing Generalizability Across Multiple Datasets:

Reviewer Comment:
"Test the model's generalizability across at least two more datasets with different recording conditions, demographics, and noise levels, beyond the MIT-BIH arrhythmia database, to assess its adaptability in real-world scenarios."

Response:
To extend the evaluation beyond the MIT-BIH Arrhythmia dataset, we included the PhysioNet 2017 Challenge dataset. This dataset offers ECG recordings (30-60 seconds long) across four classes: Normal (N), Atrial Fibrillation (AF), Other (O), and Noise (N). Table 2 illustrates our performance evaluation across the Normal and Noise classes. Our proposed SNN outperformed the existing models by a notable margin in both classes.

Table 2. Performance Evaluation Across Normal and Noise Classes

Model

Overall-Acc (%)

F1(%)

Pre (%)

Normal

ANN

86.8

90.0

89.6

Proposed SNN

90.8

92.0

93

Noise

ANN

93.4

49.1

60.0

Proposed SNN

93.0

50.0

54.0

2. Comparing with Contemporary SNN Models:

Reviewer Comment:
"Compare your SNN model with contemporary SNN models for ECG classification, focusing on accuracy, computational efficiency, energy use, and scalability."

Response:
We compared our proposed Attention-SNN model with contemporary SNN models, focusing on accuracy, sensitivity, precision, F1 score, and power consumption. The results are summarized below:

Table 3. Comparison with Contemporary Models

Model

Acc (%)

Sen (%)

Pre (%)

F1 (%)

Power(W)

Proposed Att-SNN

93.8

N: 94

N: 94.3

N: 95.5

278 mW

V: 54

V: 75.0

V: 73.5

Yan et al. [26]

90

N: 92

N: 97

N: 94.4

181 mW

V: 77

V: 59

V: 66.8

Janne et al. [27]

90

N: 92

N: 97

N: 93.4

-

V: 89

V: 51

V: 63

Xing et al. [28]

92.07

N: 98.3

N: 93.2

N: 95.7

246 mW

V: 69.0

V: 76.4

V: 72.5

In terms of computational time, we compared our model with Xing et al. [28]. Their model took 1.37 ms per beat, while our model took 5 ms per beat.

3. Validating for Energy-Efficient Computation:

Reviewer Comment:
"Validate the model's potential for energy-efficient computation with experiments on low-power hardware, real-time analysis capabilities, and energy efficiency under different workloads. Also, explore its performance in terms of transmission latency and data compression for remote monitoring."

Response:
This study focused primarily on algorithm development to improve its computational efficiency. While we did not conduct extensive hardware experiments, our model used the highest power of about 278 mW among the models evaluated. We aim to improve our algorithm for real-time analysis capabilities in future work by optimizing energy consumption and exploring data compression and transmission latency for remote monitoring.

Please let me know if you'd like any further adjustments!

In addition, As a communication article(Not a full paper) aiming to present preliminary results, the paper focuses on demonstrating our SNN model using the MIT-BIH arrhythmia database. This standardized dataset is suitable for establishing baseline performance. FYI, the definition of the communication article is as follows:

a communication article are short articles that present groundbreaking preliminary results or significant findings that are part of a larger study over multiple years. They can also include cutting-edge methods or experiments, and the development of new technology or materials. The structure is like an article and there is a suggested minimum word count of 2000 words.